# Transparent and Flexible Actuator Based on a Hybrid Dielectric Layer of Wavy Polymer and Dielectric Fluid Mixture

**DOI:** 10.3390/polym16020188

**Published:** 2024-01-08

**Authors:** Mallappa Mahanthappa, Hyun-U Ko, Sang-Youn Kim

**Affiliations:** Advanced Research Technology Center, Korea University of Technology and Education, 1600, Chungjeol-ro, Byeongcheon-myeon, Dongnam-gu, Cheonan-si 31253, Chungchungnam-do, Republic of Korea; mallappa.m@koreatech.ac.kr (M.M.); lostmago@koreatech.ac.kr (H.-U.K.)

**Keywords:** vibrotactile, transparent and flexible actuator, dielectric fluid, propylene carbonate, acetyl tributyl citrate

## Abstract

Transparent and flexible vibrotactile actuators play an essential role in human–machine interaction applications by providing mechanical stimulations that can effectively convey haptic sensations. In the present study, we fabricated an electroactive, flexible, and transparent vibrotactile actuator with a dielectric layer including a dielectric elastomer and dielectric fluid mixture. The dielectric fluid mixture of propylene carbonate (PC) and acetyl tributyl citrate (ATBC) was injected to obtain a transparent dielectric layer. To further improve the haptic performance, different weight ratios of dielectric fluid (PC: ATBC) were injected. The fabricated vibrotactile actuators based on a transparent dielectric layer were investigated for their electrical and electromechanical behavior. The proposed actuators generate a large vibrational intensity (~2.5 g) in the range of 200–250 Hz. Hence, the proposed actuators open up a new class of vibrotactile actuators for possible use in various domains, including robotics, smart textiles, teleoperation, and the metaverse.

## 1. Introduction

Beyond the foldable smartphone, lightweight mobile devices such as laptops with flexible displays are competitively announced to attract the interest of potential users. Although the devices appeal to their future-oriented characters and design, they are lacking compared to traditional devices in terms of tactile user experience, for example, typing the keyboard due to the absence of a rigid interface. To this point, vibrotactile haptic actuators have been considered a candidate to apply tactile user experience [1].

Vibrotactile haptic actuators are a kind of haptic actuators that can evoke vibration stimuli in the human skin for inducing haptic feedback in human–machine interactions. For generating vibrational stimuli, a wide variety of actuators, such as eccentric motor-based tactile actuators, linear resonance actuators, and piezoelectric transducers, are commercially used [2,3,4]. However, the actuators are not applicable for flexible devices due to their rigid body. So, the flexible actuators based on electroactive polymers (EAPs) are highlighted for flexible devices [5,6,7,8].

The EAPs are polymers that are deformed by responding to electrical stimuli [9]. Since they convert the electrical signal to a mechanical reaction without external rigid structures, they are investigated for the material of flexible actuators [10,11,12]. Stalbaum et al. reported a soft robot based on an ionic polymer metal composite [10]. Kim et al. proposed an eco-friendly electroactive paper (EAPap)-based actuator [11]. Choi et al. suggested a soft lens based on PVC gel [12].

Among the EAPs, dielectric elastomers (DEs) are widely investigated for vibrotactile actuators due to their fast response and large deformation [13,14]. Ozsecen et al. reported the vibrotactile display based on DEs in 2010 [13]. Heo et al. designed a haptic actuator based on a wavy shape to improve actuation [14]. However, the low transparency due to the diffuse reflection by complex structures limits the application of the actuator for flexible display. To increase the transparency by removing the diffuse reflection effect, adding a dielectric fluid with a similar refractive index might be one option. A hydraulically amplified self-healing electrostatic (HESEL) actuator is a kind of actuator that achieves high transparency and high performance by adding dielectric fluid [15]. However, the actuator is not applicable for a haptic actuator flexible display due to its low actuation frequency (<100 Hz) and curved shape.

In this paper, we proposed a flexible and transparent actuator with a dielectric layer including a wavy elastomer and dielectric fluid mixture of propylene carbonate (PC) and acetyl tributyl citrate (ATBC). In this mixture, PC plays the role of the high dielectric liquid. ATBC was used as an additive for increasing the dielectric short strength. The dielectric behavior of the dielectric layer was investigated using analysis of capacitance. The transparency of the dielectric layer was around 80%. The optimal concentration ratio of the dielectric mixture was defined from the vibrotactile behavior of actuators with them.

## 2. Experimental Section

### 2.1. Materials and Reagents

For the fabrication of the dielectric layer, polydimethylsiloxane (PDMS, Sylgard184, Dow Corning Corp., Midland, TX, USA), Ecoflex (Ecoflex-0010™, Smooth-On Inc., Easton, PA, USA), propylene carbonate anhydrous (PC, 99.7%, Sigmaldrich, St. Louis, MO, USA), and acetyl tributyl citrate (ATBC, 28–30%, Merck, Rahway, NJ, USA) were used.

### 2.2. Instrumentation

The UV-visible spectrometer (Cary 60, Agilent Technology, Santa Clara, CA, USA) was used to evaluate transparency. The capacitance of the dielectric layer was investigated with an impedance and phase analyzer (SI-1296, AMETEK Scientific Instruments, Oak Ridge, TN, USA). For analyzing vibration performance, the input signal was generated with a function generator (9310, Protek, Gwangmyeong-si, Gyeonggi-do, South Korea) and amplified with a high-voltage amplifier (Trek 10/40A, Denver, CO, USA). The vibration signal was corrected by an accelerometer (Type 4393, Brüel & Kjær, Nærum, Denmark) with a signal amplifier (NEXUS, Brüel & Kjær, Nærum, Denmark). The acceleration signal was displayed and recorded on an oscilloscope (TDS 1012B, Tektronix, Beaverton, OR, USA).

### 2.3. Fabrication of Dielectric Wavy Polymer

PDMS-Ecoflex wavy dielectric elastomer was fabricated by a simple molding method. In this procedure, the polydimethylsiloxane and Ecoflex solution were placed into a beaker and then stirred for 10 min. The resulting solution was poured carefully onto a wavy-patterned Teflon dish, and this was placed in a vacuum chamber to eliminate any bubbles in the mixture. Finally, the mixture was kept for drying in an electric oven at 100 °C for 3 h, and then, it was allowed to cool down to ambient temperature to obtain a PDMS-Ecoflex wavy-shaped layer. The resulting layer detaches from the Teflon dish and is used for further study.

### 2.4. Fabrication of Dielectric Fluid Mixture

PC-ATBC dielectric fluid was prepared by mixing a definite amount of polycarbonate and ATBC. The various weight ratios of PC to ATBC (P:A) were 10:0, 9.5:0.5, 9:1, 8:2, 7:3, 6:4, 5:5, and 0:10 and are denoted as PC, PA_9.5:0.5_, PA_9:1_, PA_8:2_, PA_7:3_, PA_6:4_, PA_5:5_, and ATBC, respectively (as shown in Table 1).

### 2.5. Design of Vibrotactile Actuator

Figure 1a represents the fabrication of a soft and transparent vibrotactile actuator based on a dielectric elastomer. The proposed actuator was composed of a top layer, a dielectric layer, an adhesive pillar, and a bottom layer. The top and bottom layers were made up of indium tin oxide/poly(ethylene terephthalate) (ITO/PET), which is an actuation component, and a flexible supporter. The top layer acts as a cathode, whereas the bottom layer acts as an anode. Then, a PDMS-Ecoflex-based wavy dielectric polymer layer is placed between two electrodes (top and bottom layer) and is referred to as a dielectric layer. Using VHB tape, the edges of the topmost layer are connected to the edges of the bottom layer, which is called the adhesive layer. The dielectric fluid (PC: ATBC) was injected through VHB tape using a syringe, and the wavy-shaped dielectric layer disappeared and obtained a transparent layer (as shown in Figure 1). The sizes of the top and bottom layers are 50 mm (W) × 36 mm (L) × 100 μm (T), whereas the size of the dielectric layer is 23 mm (L) × 23 mm (W) × 1.2 mm (T). Figure 1c shows the flexibility of the actuator. The proposed actuator is highly flexible and transparent; hence, it is used in haptic performance. To improve the transparency of the haptic performance, the dielectric fluid weight ratio is optimized. The haptic performance of the designed actuator greatly depends on the mechanical and dielectric properties of the dielectric layer including elastomer and dielectric liquid. To achieve the best haptic performance, it is necessary to optimize the mechanical properties of the dielectric layer. According to our previous report, the highest vibration performance of the dielectric layer was observed at the weight ratio of PDMS to Ecoflex is 1:5 [14]; hence, it was selected for further studies. To further improve the transparency and vibration performance of the dielectric layer, dielectric fluid with different weight ratios of PC to ATBC was injected. The codes of the actuator are denoted in Table 1.

### 2.6. The Vibrotactile Actuator Performance Evaluation

The actuation performance was investigated by using the accelerometer. The input signal was generated by the function generator and amplified by the high-voltage amplifier. The signal and ground line from the high-voltage amplifier were connected to the bottom and top layers as electrodes, respectively. The accelerometer was located on the top of the vibrotactile actuator. The acceleration signal was amplified by the signal amplifier and collected, displayed, and saved by an oscilloscope. To evaluate the durability of the actuator, 10,000 cycle actuation test was also conducted at normal state and then after 100 times of bending according to ASTM D790 [16].

## 3. Results and Discussion

### 3.1. Transparency of Dielectric Layers

Prior to the haptic performance, the optical transmittance of the proposed actuators was measured using UV-visible spectroscopy. Figure 2 represents the optical transmittance spectra of the dielectric samples in the absence and presence of dielectric fluid. In the absence of dielectric fluid, the dielectric layer exhibited an optical transmittance of about 40% due to the diffuse reflection by the wavy structure. In the presence of PC and ATBC, the dielectric layer is clearly visible (transparent) and exhibited 84.18% and 87.42% of optical transmittance, respectively. The dramatic improvement is due to liquid with a similar refractive index (PC: 1.42, ATBC:1.44, PDMS: 1.41) covering the wavy structure. In addition, in the presence of dielectric fluid with different weight ratios of PC to ATBC, the dielectric layer exhibited 83.95–86.24% of the optical transmittance observed in the visible region. Although the transparency is comparably lower than the dielectric layer, including the flat polymer and PC, it is sufficiently high for a transparent actuator. The obtained results are summarized in Table 2. Hence, these results suggest that the prepared dielectric samples have high optical transmittance and are suitable for applications in flexible and deformable displays.

### 3.2. Capacitance of the Dielectric Layer

To understand the behavior of the dielectric layer, capacitance-frequency measurements were performed over a frequency range of 10–10^6^ Hz, with an AC amplitude of 1 V. Figure 3 depicts the capacitance values for the vibrotactile actuators samples in the absence and presence of dielectric fluid. The capacitance value decreases with increasing frequency for all samples. This feature is commonly observed in propylene carbonate with high purity due to the mobility of ions and interfacial change [17]. At low frequencies, the movement of free anions is directed towards the anode, while free cations move towards the cathode. As a result, oppositely charged ions accumulate at the top and bottom layers. This accumulation leads to an electrostatic attraction between the two surfaces. As the frequency of the AC signal increases, the ion mobility imposes limitations on the polarization response time. At the time, fewer ions adsorbed on the top and bottom layer led to a noticeable decrease in the capacitance value. The gradual decrease in the capacitance at high frequencies is considered normal for capacitors due to the dipolar relaxation of the ions in the dielectric layer. In addition, donor–acceptor interactions on the interface of a solid surface and an organic liquid lead to the generation of surface charge and counterions in the liquid [18]. The surface charge increases the capacitance of the material.

When comparing the capacitance values of the wavy-shaped dielectric layer in the absence and presence of ATBC fluid, the capacitance is increased by around two times. However, the capacitance of the W_PC dielectric layer at 10 Hz frequency is four orders higher than the capacitance of W_ATBC due to the high polarity of PC, which leads to improved ion mobility. The capacitance of W_PC is also four times higher than F_PC, caused by the increase in the surface charge by an increase in the surface area. The data show that the actuator with W_PC will perform drastically higher actuation than actuators with W_ATBC and F_PC according to the following equation.
(1)$$a=\frac{CV^{2}}{md}$$

Here, a, C, V, m, and d are acceleration, capacitance, input voltage, mass, and thickness of the dielectric layer, respectively. However, the high capacitance probably leads to an easy electrical short [19]. So, we propose a dielectric liquid mixture of PC and ATBC in this paper to increase electric stability. The capacitance of dielectric layers including low content of ATBC as W_PA_9.5:0.5_ and W_PA_9:1_ is similar to W_PC. As the content of ATBC increases over the ratio, however, the capacitance decreases. From the result, it is estimated that the highest actuation performance might be observed at the W_PA_9.5:0.5_ and W_PA_9:1_.

### 3.3. Haptic Performance of Proposed Vibrotactile Actuators

Further, the acceleration or amplitude or vibration of a dielectric layer of the as-designed actuator was analyzed by acceleration–-frequency and acceleration-–voltage measurements. In the first experiment, the frequency response of the proposed actuator was performed by sweeping the frequency of the input signal from 0 to 300 Hz using a fixed peak-to-peak input voltage of 2 kV except for W_PC. Since the W_PC was short at the 2 kV, it was measured under a 1 kV input voltage. The W_PA_9.5:0.5_ and F_PC cases also measured under 1 kV input voltage. The vibrational acceleration of the designed actuator in the absence and presence of dielectric fluid is shown in Figure 4. In the presence of ATBC and PC, the dielectric layer shows negligible and maximum changes in the vibrational amplitude of the proposed actuator. Similarly, increasing the weight ratio of ATBC decreases the vibrational amplitude of the dielectric layer over the frequency range from 200 to 250 Hz as a function of applied frequency due to capacitance variation and the damping effect from the viscosity variation (PC: 2.0 cP, ATBC: 20 cP). The damping effect is also the reason for the broadened resonance peak, whereas the peak frequency shows an increasing trend as the increasing content of ATBC might be due to a lower density of ATBC (1.04 g/L) than PC (1.20 g/L). However, the actuator with a high content of PC shows an easy short at a lower voltage. Moreover, the actuator with a wavy shape of the polymer in the dielectric layer shows a higher vibrational amplitude than the actuator with a flat polymer due to the capacitance variation.

Additionally, the haptic performance of an actuator is influenced not only by its frequency response but also by the amplitude of the input voltage. While the frequency response determines how well the actuator can reproduce different vibration frequencies, the amplitude of the input voltage plays a crucial role in determining the strength or intensity of the haptic feedback. In a second experiment, Figure 5a shows that the haptic measurements were performed by varying the amplitude of the input voltage at the resonant frequencies of actuators. The vibration strength is directly related to the amplitude of the input voltage. At the low input voltage under 1.5 kV, W_PC shows the highest vibration due to high capacitance according to Equation (1). But it was short under 2 kV input voltage with 200 Hz frequency. On the other hand, the W_ATBC bears the highest input voltage over 3 kV at a higher frequency (250 Hz). The bearable input voltage increases as the content of ATBC increases. Even though the resonance frequency used for actuation increases depending on the increasing content of ATBC, the same trend is shown. Considering the proportional relation between frequency and AC conductivity, the result suggests the enhancement of stability by adding ATBC. By the combination of capacitance and electric stability, the actuator with a dielectric fluid of weight ratio PA_9:1_ exhibits the highest performance among all the prepared vibrotactile actuators under 2.5 kV input voltage.

## 4. Conclusions

In summary, the fabrication of a thin, flexible, transparent, and lightweight vibrotactile actuator based on a transparent dielectric layer including dielectric elastomer and dielectric fluid mixture. The proposed vibrotactile actuator is composed of a top layer, an adhesive layer, a wavy-shaped dielectric layer, and a bottom layer. To obtain a transparency of the dielectric layer, the dielectric fluid mixture of PC and ATBC was injected. By optimizing the various dielectric fluid weight ratios of PC:ATBC, a better vibrotactile performance was determined. The fabricated vibrotactile actuator-based transparent dielectric layers were subjected to capacitance, transparency, resonance frequency, and vibration strength measurements. The results obtained from the electrical measurement, the high capacitance value at low frequencies, and the gradual decrease in the capacitance value at high frequencies indicate the proposed dielectric layer is a suitable choice for actuators with a large deformation. Additionally, the proposed actuator produces large vibrational intensity in the range of 200–250 Hz and a high vibrational strength at lower applied voltages. Among the different weight ratios of PC:ATBC, PA_9:1_ exhibited the maximum vibrational intensity at 200 Hz and vibrational strength at 2.5 g. Therefore, the proposed haptic actuator is set to become a fundamental component in the future of smart devices. With its advanced tactile feedback capabilities, this technology will revolutionize the user experience across a range of devices.

## Figures and Tables

**Figure 1 polymers-16-00188-f001:**
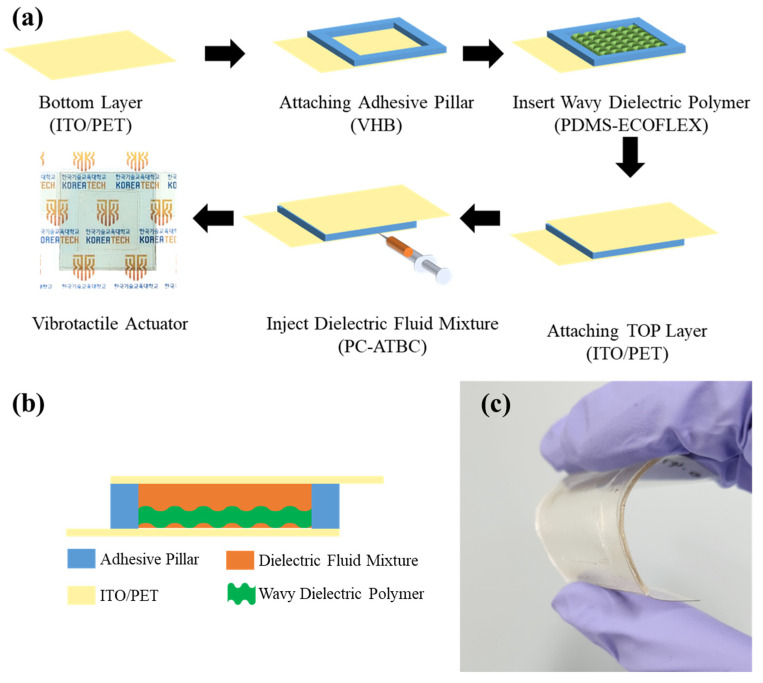
(**a**) Fabrication process, (**b**) cross-section view, and (**c**) bent image of the vibrotactile actuator.

**Figure 2 polymers-16-00188-f002:**
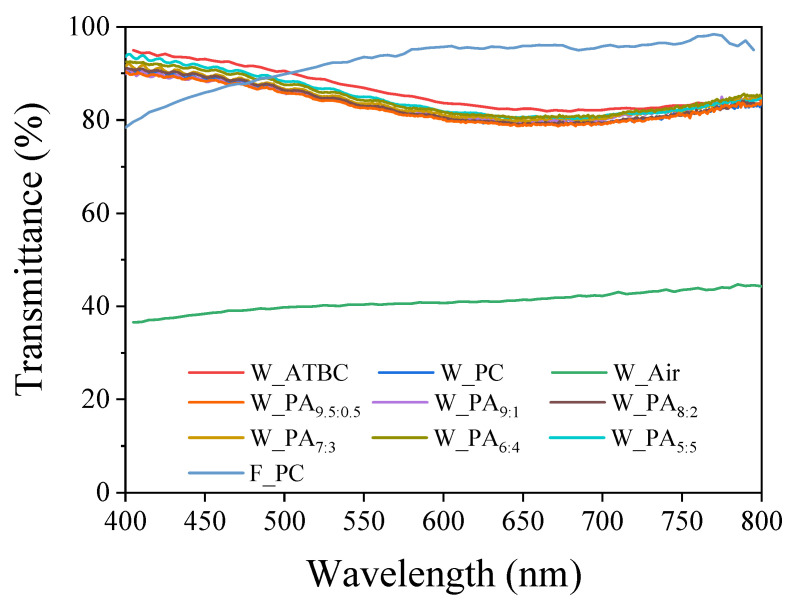
Transmittance of actuators.

**Figure 3 polymers-16-00188-f003:**
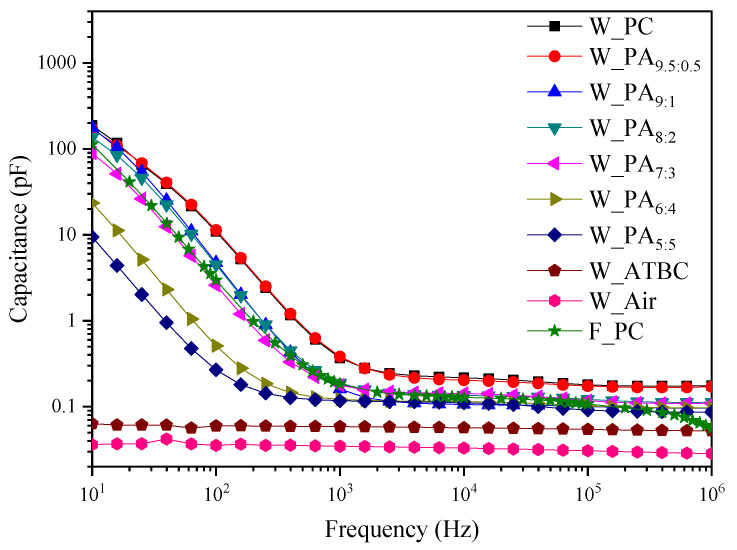
Capacitance of actuators.

**Figure 4 polymers-16-00188-f004:**
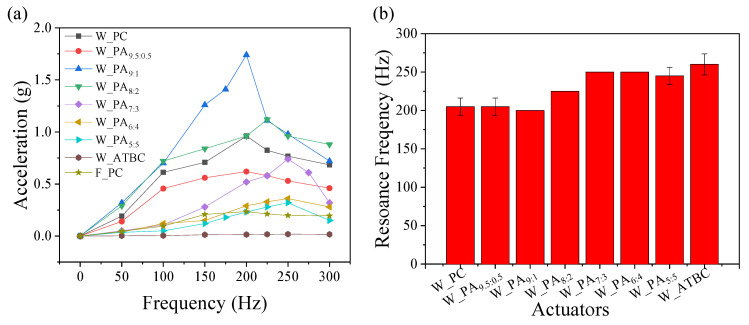
(**a**) Vibration acceleration of vibrotactile actuator under various input frequencies and (**b**) the averaged resonance frequencies for actuators.

**Figure 5 polymers-16-00188-f005:**
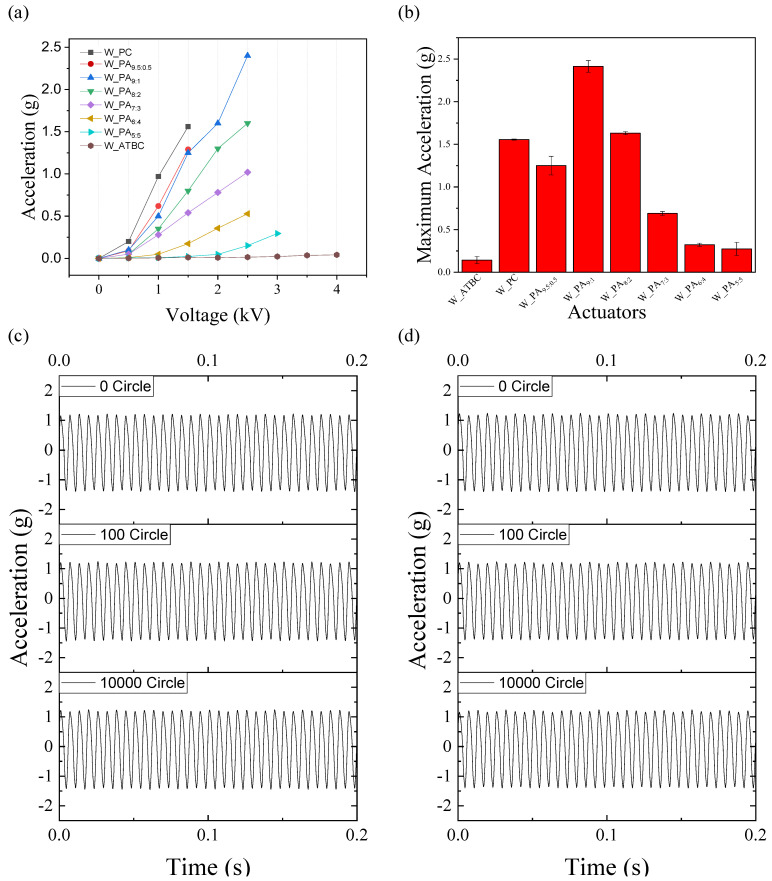
(**a**) Vibration acceleration of vibrotactile actuator under various input voltage, (**b**) maximum acceleration of various actuators, and 10,000 cycle actuation test result (**c**) at normal state and (**d**) after 100 times bending.

**Table 1 polymers-16-00188-t001:** Code of actuator.

Code	Polymer	Dielectric Liquid Ratio	
		PC	ATBC
B_Air	-	-	-
F_PC	Flat	10	0
W_Air	Wavy	-	-
W_PC	Wavy	10	0
W_PA_9.5:0.5_	Wavy	9.5	0.5
W_PA_9:1_	Wavy	9	1
W_PA_8:2_	Wavy	8	2
W_PA_7:3_	Wavy	7	3
W_PA_6:4_	Wavy	6	4
W_PA_5:5_	Wavy	5	5
W_ATBC	Wavy	0	10

**Table 2 polymers-16-00188-t002:** Averaged transmittance of the actuator.

Sample	T% _avg._
W_ATBC	87.43
W_Air	40.18
W_PC	84.18
W_PA_9.5:0.5_	83.95
W_PA_9:1_	84.39
W_PA_8:2_	84.46
W_PA_7:3_	85.01
W_PA_6:4_	85.77
W_PA_5:5_	86.24
F_PC	92.59

## Data Availability

Data are contained within this manuscript.

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
