# Peer review of "Transparent and Flexible Actuator Based on a Hybrid Dielectric Layer of Wavy Polymer and Dielectric Fluid Mixture"

_polymers, 2024, doi:10.3390/polym16020188_

Round 1

Reviewer 1 Report

Comments and Suggestions for Authors

In this paper, the authors fabricated an electroactive, flexible and transparent actuator with a dielectric layer in the middle. This paper is well structured, data results clearly presented. I would recommend this paper published after minor issues:

1. Fig 2, the W_Air transparency is only 40%. Please add more explanation.

2. Fig. 3, most of the curves have a huge increas at low frequencies. This indicates that the ions play a huge role at low frequencies. There are many impurity ions in the liquids. Maybe try purify the solutions?

Author Response

We like to thank Reviewer 1 for their valuable comments to improve the quality of the manuscript. We have addressed all the comments raised by the reviewers, with necessary explanations and experimental details. As requested a point-by-point response to the reviewer’s comments along with a detailed description of changes made is provided below.

Comment 1. Fig 2, the W_Air transparency is only 40%. Please add more explanation.

Response: Thank you for your valuable suggestions and comments. Since deflection by the wavy structure, the transparency was very low. After the dielectric fluid mixture covers the wavy structure, Transparency is dramatically improved. The exploration is incorporated in the revised manuscript (Line no. 159-160).

Comment 2. Fig. 3, most of the curves have a huge increase at low frequencies. This indicates that the ions play a huge role at low frequencies. There are many impurity ions in the liquids. Maybe try to purify the solutions?

Response: The dielectric trend is normally observed for dipolar liquid such as propylene carbonate (PC), even if it is high purity, about 99% [1]. In this paper, we use 99.99% PC. So, we think purity is not the critical point for the phenomenon. The dielectric behavior at low frequency is not only influenced by the number of ions and the interfacial shape to induce surface charge. In this paper, the dipole polarity of the PC and the shape of the polymer might be influenced by high capacitance at low frequency according to comparing with W_PC, W_ATBC, and F_PC (It is additionally tested and included in the paper). The detailed exploration is added in Lines 168 to 171.

[1] Choudhary, S.; Dhatarwal, P.; Sengwa, R.J. Characterization of conductivity relaxation processes induced by charge dynamics and hydrogen-bond molecular interactions in binary mixtures of propylene carbonate with acetonitrile. J. Mol. Liq. 2017, 231, 491-498.

Reviewer 2 Report

Comments and Suggestions for Authors

The authors use a wavy polymer structure and dielectric fluid to create a haptic actuator. Different dielectric fluids are studied and compared, and the performance of the haptic actuator is investigated. After reviewing this manuscript, a couple of issues need to be addressed before publication.

1.      The effect of the wavy structure should be studied. The authors should compare their results with a control group to demonstrate the improvement of using a wavy structure. For example, the authors can compare their results with a flat surface.

2.      A cross-sectional view of the device should be added to Figure 1. So readers can understand the structure and measuring mechanism better. Detailed dimensions of the haptic actuator should also be explained.

3.      The nomenclature of different layers in the device should be the same in text and figures. It is not easy to understand the components in the current format.

4.      The experimental setup needs to be detailed. For example, how the acceleration was measured.

5.      Major issue:

In Figure 4, it appears that there are some structure resonant effects in this haptic actuator. The contribution of the wavy structure and the dielectric filler should be discussed.

Also, the trend of the measured acceleration is inconsistent with the PA ratio. Please explain this mechanism. Can this result be repeated? Please include an error bar to demonstrate the performance of this device.

Comments on the Quality of English Language

Moderate editing of English language required

Author Response

We like to thank reviewer 2 for their valuable comments to improve the quality of the manuscript. We have addressed all the comments raised by the reviewers, with necessary explanations and experimental details. As requested a point-by-point response to the reviewer’s comments along with a detailed description of changes made is provided below.

Reviewer 2

Comment 1. The effect of the wavy structure should be studied. The authors should compare their results with a control group to demonstrate the improvement of using a wavy structure. For example, the authors can compare their results with a flat surface.

Response: Thank you for your valuable comments and suggestions. As per your suggestion, the actuator with flat elastomer was fabricated and its test results are included in the revised manuscript.

Comment 2. A cross-sectional view of the device should be added to Figure 1. So readers can understand the structure and measuring mechanism better. Detailed dimensions of the haptic actuator should also be explained.

Response: The cross-section view is included in the revised manuscript in Figure 1.

Comment 3. The nomenclature of different layers in the device should be the same in text and figures. It is not easy to understand the components in the current format.

Response: The nomenclature of different layers in the device in text and figures has been corrected in the revised manuscript.

Comment 4. The experimental setup needs to be detailed. For example, how the acceleration was measured.

Response: As per the reviewer’s suggestion, the experimental section is described in detail in the revised manuscript (Line No. 129-132).

Comment 5. Major issue: In Figure 4, it appears that there are some structure resonant effects in this haptic actuator. The contribution of the wavy structure and the dielectric filler should be discussed.

Response: According to your comment, we tested the actuator with a flat structure polymer as shown in modified Figure 4. From the result, the wavy structure is not significantly affected by the resonance frequency. it might be due to the low stiffness of silicone-based polymer. Since the similar density of propylene carbonate (PC:1.2g/cm3) and ATBC (1.05g/cm3), the dielectric filler difference also does not significantly affect the resonance frequency variation. Hence, there are similar resonance frequencies and it is explained in the revised manuscript (Line no. 202-208).

Comment 6. Also, the trend of the measured acceleration is inconsistent with the PA ratio. Please explain this mechanism. Can this result be repeated? Please include an error bar to demonstrate the performance of this device.

Response: As per the reviewer’s suggestion, the acceleration of W_PA9.5:0.5 in Figure 4 is inconsistent with the trend of acceleration since it is also investigated under 1kV input voltage as same with PC due to short phenomenon as shown in Figure 5. It has been included in the revised manuscript (Line no. 177-182).

Also, the repeat test was conducted according to your comment. However, it is too complex to include in performance result graphs. Hence, we are separately adding the maximum acceleration graph with the error bar in Figure 5.

Reviewer 3 Report

Comments and Suggestions for Authors

The paper submitted by Mahanthappa et al. describes a transparent actuator constructed by dielectric elastomer injected with PC and ATBC. The goal of this paper is not clearly defined and it lacks scientific soundness regarding many statements and also lacks scientific significance. Therefore I believe major change is necessary. 

1. In the Introduction part, the authors list some examples of EPA-based actuators. This is not sufficient for an introduction. The authors should discuss the state-of-art development of actuators using the similar configuration as used in this manuscript. What are their issues and how the invention in this manuscript solve these issues. You need to highlight your contribution and novelty. 

2. Why adding PC and ATBC can increase the transmittance?

3. Page 6 line 171, you will need to quantify the breakdown. For example, you can measurement the breakdown strength. Also, the authors mentioned “it is estimated that the highest actuation performance might be observed at the W_PA9.5:0.5 and W_PA9:1”. Still you will need data to back up this statement. 

4. How do you measure the acceleration and process the signal data?

5. Figure 4, why the acceleration shows a peak at 200-250 Hz?

6. Have you measured the lifetime or any fatigue of the actuator? For example, do you see any decay of performance after 10000 cycles. 

7. Have you measured the performance after stretching and/or bending the actuator for certain cycles?

Just a general remark (not a question to be addressed in this manuscript), application of dielectric elastomer in actuator faces big challenges such as slow response and high voltage. The authors may considering solving these problems in future studies. 

Author Response

Dear Reviewer 3,

We like to thank  Reviewer 3 for their valuable comments to improve the quality of the manuscript. We have addressed all the comments raised by the reviewers, with necessary explanations and experimental details. As requested a point-by-point response to the reviewer’s comments along with a detailed description of changes made is provided below.

Reviewer 3

Comment 1. In the Introduction part, the authors list some examples of EPA-based actuators. This is not sufficient for an introduction. The authors should discuss the state-of-the-art development of actuators using a similar configuration as used in this manuscript. What are their issues and how does the invention in this manuscript solve these issues You need to highlight your contribution and novelty. 

Response: Thanks for your valuable suggestions and comments. As per the reviewer’s suggestion, the introduction of the manuscript has been improved in the appropriate section.

Dielectric elastomers are extensively studied for vibrotactile actuators because of their remarkable attributes such as fast response and significant deformation. However, the actuator's application for flexible display is limited by the low transparency caused by complex structures, which results in diffuse reflection. One of method is elimination of diffuse reflection through the addition of a liquid with a refractive index similar to the material like hydraulic amplification self-repairing electrostatic (HASEL) actuator. Nevertheless, the HASEL actuator's low actuation frequency (<100 Hz) and curvature shape make it unsuitable for a haptic performance flexible display. In this context, we proposed a plate-shape, flexible and transparent actuator with a dielectric layer including a wavy elastomer and dielectric fluid a mixture of propylene carbonate (which serves as a high dielectric liquid) and acetyl tributyl citrate (an additive that enhances the dielectric short strength). The dielectric layer exhibits approximately 80% transparency, a significant capacitance value at low frequencies, and a large vibrational intensity (~2.5 g) within the 200-250 Hz range. Thus, the proposed haptic actuator is poised to become an essential element in the forthcoming era of smart devices.

Comment 2. Why adding PC and ATBC can increase the transmittance?

Response: Because the organic liquids with similar refractive index (PC: 1.42, ATBC: 1.44, PDMS: 1.41, Ecoflex: 1.40 according to the datasheet from the maker) and dielectric polymer covering the wavy shape, which was caused by the diffuse reflection.

Comment 3. Page 6 line 171, you will need to quantify the breakdown. For example, you can measure the breakdown strength. Also, the authors mentioned, “It is estimated that the highest actuation performance might be observed at the W_PA9.5:0.5 and W_PA9:1”. Still, you will need data to back up this statement. 

Response: In this paper, the breakdown strength was not investigated. For clear exploration, we will change the “breakdown” to “short”.

Comment 4. How do you measure the acceleration and process the signal data?

Response: The detailed procedure has been included in the revised manuscript.

Comment 5. Figure 4, why does the acceleration show a peak at 200-250 Hz?

Response: It might be due to the mechanical properties of the Top electrode materials and their structure. From the result of the actuator with a flat polymer structure as shown in modified Figure 4, the wavy structure is not significantly affected by the resonance frequency. It might be due to the low stiffness of silicone-based polymer. Since the similar density of propylene carbonate (PC:1.2g/cm3) and ATBC (1.05g/cm3), the dielectric filler difference also does not significantly affect the resonance frequency variation. Hence, there are similar resonance frequencies.

Comment 6. Have you measured the lifetime or any fatigue of the actuator? For example, do you see any decay of performance after 10000 cycles? 

Response: In this paper, we are mainly focused on increasing transparency. Thus, the lifetime and fatigue are not considered.

Comment 7. Have you measured the performance after stretching and/or bending the actuator for certain cycles?

Response: In this paper, we majorly emphasize increasing transmittance by adding dielectric fluid, however, bending durability is not considered.

Round 2

Reviewer 2 Report

Comments and Suggestions for Authors

The authors have replied to my comments with a revision. Most comments were properly replied to except for the two major issues in comments 4 and 5. Therefore, I cannot recommend this paper for consideration in the current format.

The authors should really spend time studying the material, physical, and mechanical effects that contribute to this device. It is clear from the experimental data that another effect overwrites the contributions of the PA ratio. Take Figure 4 as an example, W_PA9.5:0.5 and W_PA7:3 is crossed at 225 Hz. The large peak from 150 Hz to 220 Hz for W-PA9:1, etc. Also, it is improper to compare the maximum acceleration in Figure 5. They should be compared in identical experimental conditions. Adding a resonant effect is okay, but it also suggests that the PA ratio is not the major amplification mechanism in this sensor.

Author Response

Dear Reviewer 2,

We like to thank Reviewer 2 for their valuable comment to improve the quality of the manuscript. We have addressed the comment raised by the reviewer 2, with necessary explanation and data. As requested response to the reviewer 2’s comment along with a detailed description of change made is provided below.

Reviewer 2

Comment 1. The authors should really spend time studying the material, physical, and mechanical effects that contribute to this device. It is clear from the experimental data that another effect overwrites the contributions of the PA ratio. Take Figure 4 as an example, W_PA9.5:0.5 and W_PA7:3 is crossed at 225 Hz. The large peak from 150 Hz to 220 Hz for W-PA9:1, etc. Also, it is improper to compare the maximum acceleration in Figure 5. They should be compared in identical experimental conditions. Adding a resonant effect is okay, but it also suggests that the PA ratio is not the major amplification mechanism in this sensor.

Response: Thank you for your valuable comments and suggestions. As per your suggestion, we carefully checked the data and retested some samples such as W_PC, W_PA9.5:0.5, and W_PA5:5. According to the result as shown in revised Figure 5 the resonance frequency shows a decreasing trend as increasing content of PC. It might be due to 20% higher density of PC (1.20g/l) than ATBC (1.04g/l). The modified descript is added at Line 213 on Page 7. Figure 6 was also revised according to the test result. 

Reviewer 3 Report

Comments and Suggestions for Authors

Some of my questions are not adequately addressed here. Nevertheless, I'll give an ok to this manuscript.

Author Response

Dear Reviewer 3,

We like to thank Reviewer 3 for their valuable comment to improve the quality of the manuscript. We have addressed the comment raised by the reviewer 3, with necessary explanation and data. As requested response to the reviewer 3’s comment along with a detailed description of change made is provided below.

Reviewer 3

Comment 1. Some of my questions are not adequately addressed here. Nevertheless, I'll give an ok to this manuscript.

Response: Thank you for your valuable comment. As per your suggestion, we conducted a 10000-cycle actuation test for the actuator at normal state and after 100 times bending. The result data and description are added to Figure 6c-d and Line 233 of Page.

Round 3

Reviewer 2 Report

Comments and Suggestions for Authors

The authors replied to the comments I raised for the revision. The authors suggest that the cross between W_PA9.5:0.5 and W_PA7:3 may be due to the increasing PC weight content. This might be the reason for this effect. The authors should use all the data to verify this statement. For example, the authors can draw a box plot of peak frequency versus PC weight ratio to discuss this effect.

As for Figure 5, the authors still need to answer my comment. The comparison among all parameters should based on the same measurement conditions. In this case, I think the authors picked resonant peaks of each sample to do this comparison. This means this plot does not reveal the contribution of the W_PA ratio. It can have a combined effect of the W_VA ratio and weight effect. Or, it could have contribution from the structure resonance. I suggest that the authors clarify the effectiveness of these different PA ratios to the readers. It can highlight the main contribution of this paper.

Author Response

Dear Reviewer 2,

We like to thank Reviewer 2 for their valuable comment to improve the quality of the manuscript. We have addressed the comment raised by reviewer 2, with the necessary explanation and data. As requested response to reviewer 2’s comment along with a detailed description of the change made is provided below.

Reviewer 2

Comment 1. The authors replied to the comments I raised for the revision. The authors suggest that the cross between W_PA9.5:0.5 and W_PA7:3 may be due to the increasing PC weight content. This might be the reason for this effect. The authors should use all the data to verify this statement. For example, the authors can draw a box plot of peak frequency versus PC weight ratio to discuss this effect.

Response: Thank you for your valuable comments and suggestions. As per your suggestion, we considered drawing the box plot. However, the data did not show enough variation to draw a box plot due to the large gap in test frequencies. So, we add the figure of the average and standard deviation of the resonance frequencies for each actuator.

Comment 2. As for Figure 5, the authors still need to answer my comment. The comparison among all parameters should based on the same measurement conditions. In this case, I think the authors picked resonant peaks of each sample to do this comparison. This means this plot does not reveal the contribution of the W_PA ratio. It can have a combined effect of the W_VA ratio and weight effect. Or, it could have contribution from the structure resonance. I suggest that the authors clarify the effectiveness of these different PA ratios to the readers. It can highlight the main contribution of this paper.

Response: We agree with your valid point. The increment of amplitude of actuation and resonance frequency is explained in detail (Figure 4a&b and line 211 on page 7). As per your suggestion, the effectiveness of different ATBCs has been clarified and highlighted (line 230 on page 7) in the revised manuscript.

Round 4

Reviewer 2 Report

Comments and Suggestions for Authors

This manuscript can be accepted for publicaiton.